# A Mouse Holder for Awake Functional Imaging in Unanesthetized Mice: Applications in ^31^P Spectroscopy, Manganese-Enhanced Magnetic Resonance Imaging Studies, and Resting-State Functional Magnetic Resonance Imaging

**DOI:** 10.3390/bios12080616

**Published:** 2022-08-08

**Authors:** Lindsay C. Fadel, Ivany V. Patel, Jonathan Romero, I-Chih Tan, Shelli R. Kesler, Vikram Rao, S. A. Amali S. Subasinghe, Russell S. Ray, Jason T. Yustein, Matthew J. Allen, Brian W. Gibson, Justin J. Verlinden, Stanley Fayn, Nicole Ruggiero, Caitlyn Ortiz, Elizabeth Hipskind, Aaron Feng, Chijindu Iheanacho, Alex Wang, Robia G. Pautler

**Affiliations:** 1Department Integrative Physiology, Baylor College of Medicine, Houston, TX 77030, USA; 2Department of Neuroscience, Baylor College of Medicine, Houston, TX 77030, USA; 3School of Humanities, Rice University, Houston, TX 77005, USA; 4Small Animal Imaging Facility, Texas Children’s Hospital, Houston, TX 77030, USA; 5Bioengineering Core, Advanced Technology Core, Baylor College of Medicine, Houston, TX 77030, USA; 6School of Nursing, University of Texas at Austin, Austin, TX 78712, USA; 7Department of Chemistry, Wayne State University, Detroit, MI 48202, USA; 8Cancer and Cell Biology Program, Baylor College of Medicine, Houston, TX 77030, USA; 9Department of Pediatrics, Texas Children’s Cancer and Hematology Centers and The Faris D. Virani Ewing, Houston, TX 77030, USA; 10Sarcoma Center, Baylor College of Medicine, Houston, TX 77030, USA; 11Department of Neuroscience, Augustana College, Rock Island, IL 61201, USA; 12School of Molecular and Cellular Biology, University of Illinois at Urbana-Champaign, Urbana, IL 61801, USA; 13Department of Radiology, Baylor College of Medicine, Houston, TX 77030, USA; 14Huffington Center on Aging, Baylor College of Medicine, Houston, TX 77030, USA

**Keywords:** MRI, mouse, animal holder, awake, MEMRI, rs-fMRI, anesthesia

## Abstract

Anesthesia is often used in preclinical imaging studies that incorporate mouse or rat models. However, multiple reports indicate that anesthesia has significant physiological impacts. Thus, there has been great interest in performing imaging studies in awake, unanesthetized animals to obtain accurate results without the confounding physiological effects of anesthesia. Here, we describe a newly designed mouse holder that is interfaceable with existing MRI systems and enables awake in vivo mouse imaging. This holder significantly reduces head movement of the awake animal compared to previously designed holders and allows for the acquisition of improved anatomical images. In addition to applications in anatomical *T*_2_-weighted magnetic resonance imaging (MRI), we also describe applications in acquiring ^31^P spectra, manganese-enhanced magnetic resonance imaging (MEMRI) transport rates and resting-state functional magnetic resonance imaging (rs-fMRI) in awake animals and describe a successful conditioning paradigm for awake imaging. These data demonstrate significant differences in ^31^P spectra, MEMRI transport rates, and rs-fMRI connectivity between anesthetized and awake animals, emphasizing the importance of performing functional studies in unanesthetized animals. Furthermore, these studies demonstrate that the mouse holder presented here is easy to construct and use, compatible with standard Bruker systems for mouse imaging, and provides rigorous results in awake mice.

## 1. Introduction

Anesthetic agents are often used in preclinical studies that incorporate mouse or rat models. However, multiple reports indicate that anesthesia has a significant impact on the physiology of animals [1,2,3]. For example, anesthetic agents, like ketamine or isoflurane can cause physiological disruptions including decreased arterial dilation and obliterated venous dilation and are, therefore, a confounding factor in studies assessing the physiology of an animal model [4]. Neuroimaging studies that incorporate functional magnetic resonance imaging (fMRI) are a prime example of this. To control motion, fMRI studies commonly use anesthetics, the most common of which is isoflurane. FMRI relies on blood oxygenation level-dependent contrast (BOLD), which measures the hemodynamic response in brains, in which local capillaries adapt to deliver oxygen to active neurons at a faster rate than inactive neurons. Isoflurane can also alter neurovascular coupling through changes in neurophysiology, interfering with cortical neural processing and suppressing vascular cell functions [4]. Isoflurane has been shown to decrease the cerebral metabolic rate of oxygen (CMRO_2_) and, therefore, suppress functional connectivity, even after a short duration of isoflurane administration (30 min). Changes in the CMRO_2_ indicate neural hyperactivity and cell death and, thus, reflect the level of neuronal viability and brain health. Consequently, the use of these anesthetics may compromise the translational potential of mouse studies to human studies [5,6,7,8].

There have been efforts to mitigate the effects of anesthesia in functional studies. Some studies have shown that medetomidine or its enantiomer, dexmedetomidine, with or without isoflurane, might minimize some of the effects of anesthesia [9,10]. Isoflurane used in conjunction with medetomidine might reduce the dose required of both individual anesthetics, decreasing the overall physiological disruptions. A study conducted by Bukhari and colleagues using dual regression analysis revealed differences in functional connectivity between isoflurane and medetomidine anesthetized mice [9]. A major limitation of using medetomidine as a primary anesthetic is that it reduces intracortical connectivity compared to using isoflurane alone. Combining isoflurane and medetomidine at low doses partially recovers the intracortical interactions observed with the use of isoflurane alone and mitigates the disruption caused by medetomidine alone. 

Although combining low doses of isoflurane and medetomidine reduces the effects of either anesthetic alone, there is compelling evidence that demonstrates the need for awake imaging. In a study consisting of multiple trials conducted by Magnuson and coworkers, increased durations of isoflurane administration prior to functional imaging under dexmedetomidine anesthesia led to prolonged suppression of functional connectivity [10]. Administration of isoflurane for short durations (30 min) in conjunction with subsequent dexmedetomidine administration led to early signs of the suppression of functional connectivity, but these effects stopped within 1.25 h after administration was terminated. After the effects of isoflurane diminish, a secondary effect on functional connectivity appears that might be attributed to the use of dexmedetomidine. Although it is difficult to distinguish between the effects of these two anesthetics, it is reasonable to assume that they both alter normal functional connectivity [10]. Optimizing the anesthetic regime is critical to performing biologically relevant studies and obtaining meaningful and accurate results; however, the results from these studies further illustrate the importance of developing a reliable way to conduct imaging in awake animals. Thus, there has been great interest in performing preclinical imaging studies in awake, unanesthetized animals, especially for assessing functional connectivity in the brain.

Here, we report a newly designed mouse holder that is interfaceable with existing MRI systems, is feasible to create with a three-dimensional printer, and enables awake in vivo mouse imaging. Additionally, we report a five-day acclimation paradigm that, when used with the new holder, stabilized stress levels of mice, enabling improved quality of anatomical T_2_-weighted images. We have also developed an acclimation chamber that can be used in place of the MRI machine to improve efficiency and reduce machine use. Furthermore, we have included several studies measuring diverse aspects of brain metabolism and function to illustrate the importance of performing awake imaging.

First, we assessed the effects of anesthesia on high energy phosphate metabolism using ^31^P NMR spectroscopy. In vivo ^31^P NMR spectra provide a noninvasive, straightforward method for obtaining ratios of phosphorous metabolites that are involved in ATP generation and use. These studies allowed us to determine differences in phosphorus metabolite levels and ratios, providing information about the energy reserve and consumption in the brain, as well as overall metabolic activity in the brain in anesthetized versus awake mice [11]. Differences in levels of these metabolites, and ratios of these metabolites to ATP, in both awake and anesthetized mice has provided valuable information about the impact of anesthesia on brain activity and function.

Second, we assessed the effects of anesthesia on axonal transport using Manganese Enhanced Magnetic Resonance Imaging (MEMRI). MEMRI is a powerful tool that can be used to assess neuronal function independent of blood flow. Mn^2+^ is a paramagnetic ion that shortens the T_1_ of tissues and, therefore, functions as an excellent T_1_ contrast agent [12]. Mn^2+^ enters neurons through voltage-gated calcium channels and the Na^+^/Ca^2+^ exchanger and is then transported along axons and crosses synapses to neighboring neurons [13]. These properties enable MEMRI to be a functional and effective tool for studying neuronal pathways and brain functional connectivity [12,13]. Previous work demonstrated that the use of isoflurane reduced the manganese transfer index, indicating that the use of this anesthetic impacts the rate of Mn^2+^ transport [14]. In this current study, the rate of axonal transport of Mn^2+^ in awake versus anesthetized mice was measured and provided further evidence supporting the effects of anesthesia on neuronal activity.

Finally, we assessed the impact of anesthesia on resting state-fMRI (rs-fMRI) readouts. Unlike fMRI, rs-fMRI does not require the presence of stimuli or for the subject to perform a task. Therefore, rs-fMRI is used to measure spontaneous low-frequency neural activity between spatially distinct regions at rest [15,16]. Our rs-fMRI studies assessed the functional network connectivity between 62 different brain regions and helped elucidate the effects of anesthesia on fMRI readouts.

Using these imaging modalities, we illustrate the importance of performing awake mouse imaging to obtain more physiologically relevant data. In order to effectively perform awake imaging, we have designed a mouse holder that is interfaceable with existing MRI systems and enables awake in vivo mouse imaging. This holder significantly reduces head movement of the awake animal, compared to previously designed holders, and allows for the acquisition of improved anatomical images as well as improved readouts from functional imaging.

## 2. Materials and Methods

All experimental protocols were approved by the institutional animal care and use committee (IACUC) at Baylor College of Medicine and were in accordance with the guidelines published in the National Institutes of Health Guide for Care and Use of Laboratory Animals.

Separate groups of mice were used for the ^31^P, MEMRI, and rs-fMRI studies to prevent confounds due to stress, such as history or practice effects, that would reduce internal validity of these studies. Each group size was determined based upon past research.

### 2.1. Mouse Holder Design

The newly designed mouse holder was inspired by the holder developed by Stenroos and colleagues [17]. Additions to this original version of the mouse holder were added because the Stenroos version was compatible for rats, rather than for mice. The final designed holder, (Appendix A), is interfaceable with, and detachable from, the ^1^H and heteronuclear surface coils and as well as with volume resonators. The newly designed mouse holder was designed with several key elements to enable awake imaging. This includes a reduced inner volume of the nose cone that includes a trimmed top to allow the surface coil to fit (Appendix A), and a mouse bed that has an indented inner surface for a mouse body to firmly fit, a longer bed for a mouse to lay on, and the incorporation of an oxygen hose through which a tooth bar was placed to reduce head motion during imaging (Appendix A). Additionally, an altered neck bridge, with a larger inner cutout (Appendix A), and the addition of a cheek supports, in order to further reduce head motion (Appendix A), are included. STL files are included in the Appendix A in order to provide easy access to this design.

To properly secure the mice in the holder, mice are placed in the dorsal, anatomical position with their heads fitted into the built-in head brace. They are placed on a tooth bar and their nose fitted snuggly into the nose cone, with pieces of gauze placed on each side of the head for increased head stability. Artificial tears are administered, a temperature probe is inserted, and a respiration monitor is placed under the mice. To ensure the mice remain properly restrained in the holder, all four feet and the tail are taped down. Additionally, a piece of gauze is placed over the animal’s body and head and tape is firmly secured around the animal to prevent any movement (Figure 1). This mouse holder was tested with mice ranging in weight from 20 g to 50 g. The STL files have been provided in the Appendix A. The size of the nose cone can be adjusted as needed to fit a variety of mouse sizes. The body of the holder and restraint methods allow for the use of the holder for any size mouse.

### 2.2. Acclimation Protocol

A total of 10 6-week-old C57BL/6 (5 male, 5 female) underwent the acclimation protocol in the MRI machine and 6–8-week old mice (3 male, 3 female) underwent the acclimation protocol in the newly designed acclimation chamber (Figure 2A,B). Mice underwent a five-day conditioning paradigm to acclimate them to the awake imaging holder and environment (Figure 3A–F). Mice were lightly anesthetized under 0.5–1% isoflurane with a flow rate of 2.5 L/minute for approximately 2 to 3 min in order to safely transfer them to the holder. Once transferred, mice were secured into the holder with a continued delivery of 0.5–1% isoflurane (securing the mice takes approximately 2 min) and maintained at 37 °C with an air-heating system. Respiration and temperature were recorded throughout the acclimation period using either an MRI compatible SA Instruments, INC system or the Kent Scientific PhysioSuite Monitoring System, which can be used with the acclimation chamber.

Once secured, isoflurane was discontinued, and mice were given 15 min to become fully awake (based on respiration levels returning to physiologically normal levels [18,19,20]. After the 15-min period, mice remained in the holder for a set amount of time: on day one of acclimation, animals spent 10 min fully awake in the holder. Ten additional minutes were added each subsequent day, so that on day five of acclimation, mice remained fully awake in the holder for 50 min. Through this schedule, mice were gradually acclimated to the imaging environment without enduring excessive stress. The animal stress levels were monitored using a small animal monitoring physiological monitoring system from Small Animal Instruments (Stony Brook, United States) throughout the acclimation phase as well as awake imaging. Temperature was monitored via anal probe, highest measured breathing rate via breathing pad, heartrate via the same measurement pad, and the number of urinations and pellets excreted after each acclimation and imaging session. A 9.4 T Bruker Advance III 20 cm bore MRI was used for all imaging.

### 2.3. Acclimation Chamber Construction

The chamber used in acclimation protocols was designed to, as accurately as possible, mimic the environment of a small animal MRI core (Figure 2). The chamber utilized a plastic box for the casing, and for the core itself, PVC pipe (length 600mm, diameter 80 mm) was used. The pipe was affixed to a platform on the bottom of the box using steel beams to both stabilize the pipe and allow it to be removed entirely from the chamber. The pipe was enclosed in the middle of the chamber (300 mm) and extended out with the top removed at the front of the chamber (300 mm). Two openings were drilled in the front and back of the chamber. The larger front opening (90 mm) was used as the open face of the core where the mouse could be placed into the animal holder and where the animal holder itself could be slid into the makeshift core. The smaller back opening was used to feed a plastic lining/tube through to deliver isoflurane to anesthetize the mouse as needed, as well as provide the necessary oxygen flow directly to the mouse throughout the awake acclimation. Two Bluetooth speakers were placed inside the chamber and used to play recordings of MRI sounds.

### 2.4. Assessment of Mouse Cortisol Levels

Whole blood samples were taken from control mice (mice that did not undergo acclimation) as a baseline measure of cortisol levels. Whole blood samples were taken from mice that underwent the acclimation protocol directly following the end of acclimation on Day 1 and Day 5. Blood samples were taken through the submandibular method, using lancets (Goldenrod, 4 mm) and then incubated for 45 min at 37 °C. The whole blood samples were then centrifuged on 4C, 8000 rpm, for 15 min to separate the serum layers. The Cayman Chemical cortisol ELISA assay was used and read under 415 nm within 5 min after performing all steps in the protocol. Results were exported onto Excel and analyzed. The mean absorbance for each set of duplicate standards were calculated, and this was plotted on the cortisol standard curve and a line of best fit was calculated to predict the concentration of cortisol via absorbance of serum samples. These mice were not used in the subsequent studies below. Analysis of sample concentration were graphed and further analyzed in Prism.

### 2.5. T_2_-Weighted MRI

The same 10 mice that underwent the 5-day acclimation in the MRI machine were then imaged while awake. The mice were lightly anesthetized under 0.5–1% with a flow rate of 2.5 L/minute isoflurane solely for the purpose of being safely placed within the holder, at which point the isoflurane was discontinued. Isoflurane was administered for a total of approximately 2–5 min prior to each acclimation and imaging session. Imaging did not commence until respiration levels had returned to physiologically normal levels, about 160 breaths per minute [18,19,20], indicating that the animal was no longer under the influence of anesthesia (approximately a 15 min period) (Figure 3A–E,G).The T_2_-weighted imaging parameters were TR (rep time) = 2500 ms, TE (echo time) = 11 ms, slice thickness = 1 mm, matrix = 256 × 256, spatial resolution = 0.156 mm/pixel, and number of averages = 1. Ten additional mice were used as the anesthetized controls (5 male, 5 female). These mice were anesthetized under 2.5% isoflurane and were kept under anesthesia for the entirety of imaging.

### 2.6. ^31^P Spectra Studies

The ^31^P studies included 5 anesthetized (3 male, 2 female) and 4 awake mice (2 male, 2 female). In ^31^P studies, mice were imaged to measure in vivo Pi, PCr, and ATP levels. Spectra were collected using an Image Selected in vivo Spectroscopy sequence with a TR = 2 s, and number of averages = 192. Free induction decay was zero-filled, and a line-broadening of 20 Hz was applied before the Fourier transformation. ^31^P spectra were analyzed with Topspin within the Paravision 5.1 software. PCr, ^31^P, and ATP levels were measured. The alpha, beta, and gamma ATP peaks were resolved and used for the ATP assessments. Unpaired *t*-tests were used in statistical analysis.

### 2.7. MEMRI Studies

Manganese was intranasally administered to 7 mice in total. Four mice (2 male, 2 female) underwent anesthetized imaging, and 3 mice (2 female, 1 male) underwent awake imaging. Additionally, 10 mice were used as controls and were not administered manganese. Five of these control mice underwent anesthetized imaging and 5 underwent awake imaging. To intranasally administer manganese, mice were anesthetized under 2.5% isoflurane and 4 μL of 1 mg/mL manganese chloride dissolved in saline was delivered (1 μL at a time in alternating nostrils). Mice undergoing imaging while anesthetized were kept under anesthesia for an additional 45 min before T_1_ weighted MRI datasets were collected. Mice undergoing awake imaging were kept under anesthesia for an additional 15 min while in a prone position and then allowed to fully become awake for 30 min under no anesthesia before T_1_ weighted MRI datasets were collected. Axonal transport measurements in the olfactory bulb of mice were obtained using a T_1_-weighted multi-spin, multi-echo (MSME) pulse sequence as the Pautler group has previously used [12,21]. Parameters for the RARE 48.384 s scan included TR = 504.000 ms, TE = 8.500 ms, a Rare Factor of 4, 1 repetition, 1 echo image, and 2 averages. In the MEMRI studies, a 32-min MSME pulse sequence was used with TR = 500.000 ms, TE = 10.477 ms, 1 average, and 15 repetitions. As described in our previous work, MRI signal intensities in each data set were measured in the olfactory bulb and then normalized to unaltered muscle within the same slice. The location of this slice was always 1 mm anterior from the posterior edge of the olfactory bulb [21]. AVONA tests were used in statistical analysis due to their appropriateness.

Anesthetized mice were kept under 0.5–1% isoflurane levels [17] and all mice were maintained at approximately 37 °C using an air-warming system during imaging. Respiration and temperature were monitored throughout imaging by placing a breathing pad underneath the animals.

### 2.8. Rs-fMRI Studies

Rs-fMRI was conducted on 11 awake mice (5 male, 6 female) and 5 anesthetized mice (2 male, 3 female). In rs-fMRI, spontaneous BOLD signal alterations were acquired in the absence of a stimulus or task [22,23]. A spin-echo based echo planar imaging (EPI) method was applied using a micro-gradient. A 12-min free induction decay signal was obtained with parameters TR = 1200 ms, TE = 13.844 ms (minimum echo time = 13.803 ms), matrix: 64 × 64 × 30, voxel: 0.25 × 0.25 × 0.6 mm, 600 repetitions, and 1 average. The T_2_-weighted imaging parameters were TR (rep time) = 2500 ms, TE (echo time) = 11 ms, slice thickness = 1 mm, matrix = 256 × 256, spatial resolution = 0.156 mm/pixel, and number of averages = 1.

A brain mask was manually delineated in 3D for the T_2_ and rs-fMRI volumes in FMRIB Software Library (FSL) View v3.2.0 to remove the skull. Rs-fMRI data were preprocessed including realignment and warping of the EPI volume via the co-registered T_2_-weighted volume to an Allen Mouse Brain template (0.05 × 0.05 × 0.05 mm) [24] in Statistical Parametric Mapping v12. Motion was estimated using framewise displacement of the six realignment parameters (X, Y, Z, pitch, roll, yaw). The average root mean square values were calculated across all volumes to detect motion outliers as defined by 2 standard deviations from the mean across all subjects and/or 0.125 mm, which is half the average in-plane resolution of the original data [25]. A cutoff of 0.125 mm was used to detect spikes across the timeseries for each subject. Subjects that had more than 10% spikes in the timeseries were to be discarded.; however, no mice in the study exceeded this threshold.

CONN Toolbox v19 software was then used to filter data to the <0.1 Hz range of spontaneous activity [26], regressing out the motion parameters from realignment and physiologic/non-neuronal artifacts using the CompCor method [27]. Functional time series were extracted from each of 62 bilateral cortical and subcortical gray matter regions of interest to (Appendix A) cover the entire brain, cross-correlated and normalized using Fisher r-to-z transformation.

### 2.9. Statistical Analysis

*Cortisol Studies*: A one-way ANOVA was used in order to assess changes in cortisol levels between naïve (unacclimated) mice and acclimated mice on day 1 and day 5.

*^31^P Spectra*: Statistical measures were obtained in ^31^P studies using an unpaired *t*-test.

*MEMRI*: MEMRI data was analyzed using a two-way ANOVA (F(3, 11) = 21.44, *p* < 0.0001). Tukey’s multiple comparison test was used to assess the differences in axonal transport rates of anesthetized vs awake mice.

*rs-fMRI*: To analyze the rs-fMRI data, the mean, unthresholded, functional connectivity matrices for awake and anesthetized mice were compared using Mantel’s test, which provides a permutation statistic that signifies the similarity between correlation matrices [28]. This statistic was calculated using 2000 permutations. Small-worldness index was calculated, given that brain networks have been shown to demonstrate a small-world organization characterized by high local connectivity (clustering) and economical long-range connectivity (path length) [29]. Small-worldness index is computed as the ratio of clustering to path length, normalized by the clustering and path length of random networks and, therefore, this metric provides insight regarding the biological plausibility of functional connectivity matrices. Small-worldness was calculated as the area under the curve (AUC) across thresholds from minimum connection density, wherein all nodes had become fully connected in the brain networks of both groups (38%) to maximum density of 50% at intervals of 1% [30]. Small-worldness index and framewise displacement between the groups, using the Wilcoxon ranksum test, were also calculated. Movement was measured by root mean squared framewise displacement.

## 3. Results

### 3.1. Cortisol Assessment

Mouse Cortisol ELISA tests were performed on centrifuged serum collected from naïve unacclimated mice in addition to mice on Day 1 and Day 5 of the acclimation regimen. The naïve mice (*n* = 10) had an average cortisol concentration of 0.4581 ng/mL. When analyzed the Day 1 acclimated mice (*n* = 6) had an average cortisol concentration of 4.074 ng/mL. When compared using a one-way ANOVA, the naïve mice and Day 1 of acclimation regimen were found to have statistically different cortisol concentrations (*p* = 0.003) (Figure 4). The experiment was also performed on serum from day 5 of the acclimation cohort and it was found that the cortisol concentration average was 1.620 ng/mL. The ANOVA analysis found that there was no significant difference in the cortisol averages on Day 5 of acclimation and the naïve unacclimated mice.

### 3.2. Assessment of Mouse Holder

The newly designed mouse holder (Appendix A), in comparison to the Stenroos design [17], has an altered neck bridge with a larger inner cutout, and a reduced inner volume of the nose cone that consists of a shaved top surface to enable the surface coil to fit, an indented inner surface, a longer bed for the mouse to lay on, and the incorporation of an oxygen hose through which a tooth bar is placed to reduce head motion [17]. Highly motion-sensitive T_2_-weighted images, attained through awake-animal imaging, demonstrated the reduction of motion artifacts when using the newly designed restraint and new acclimation procedures (Figure 5A) compared to the original holder (Figure 5B).

### 3.3. ^31^P Spectra Studies

The ^31^P studies revealed that anesthetized mice demonstrated a significant decrease in PCr (phosphocreatine) and Pi (inorganic phosphate) compared to awake mice (Figure 6A,B). However, the ATP levels remained the same between anesthetized and awake mice (Figure 6C). These data indicated that it was likely that the metabolic flux changed in anesthetized mice. Assessment of the PCr/ATP and Pi/ATP ratio in awake and anesthetized mice demonstrated that anesthetized animals had a decrease in the energy reserve in the brain, indicated by a smaller PCr/ATP ratio, as well as a decrease in energy consumption in the brain, indicated by a smaller Pi/ATP ratio (Figure 6D,E). Additionally, anesthetized mice had a decrease in the Pi/PCr ration, indicating a decrease in metabolic activity and mitochondrial function (Figure 6F). To confirm the quality of the spectra for both anesthetized and awake (Appendix A), the Signal to Noise Ratio was measured (Appendix A).

### 3.4. MEMRI Studies

Data obtained from the MEMRI studies (Figure 7) showed that the rate of axonal transport of Mn^2+^ in the olfactory bulb of anesthetized animals was 0.0161, which was consistent with what researchers in this study, and others [18], have reported in the past (0.015–0.017 signal intensity/time). The mean rate of axonal transport of Mn^2+^ in the olfactory bulb of awake animals was 0.03869. The dramatic increase in the rate of axonal transport from Mn^2+^ anesthetized to Mn^2+^ awake mice illustrated the increased axonal transport rates of awake mice, compared with anesthetized mice. Importantly, the transport of Mn^2+^ demonstrated that axonal transport rates in the olfactory bulb were almost three-fold faster in awake animals, compared with anesthetized animals.

### 3.5. Rs-fMRI Studies

The rs-fMRI studies revealed a significant increase in the functional connectivity in awake mice, compared to anesthetized mice, illustrated by the mean, unthresholded functional connectivity matrices (Figure 8A,B). These matrices were highly similar per the Mantel statistic (r = 0.61 *p* < 0.001). The boxplot of the small-worldness index AUC was significantly higher in the awake group (W = 1, *p* < 0.001) (Figure 8C). Movement was below the cutoff for exclusion (0.06 mm) for all mice and did not differ significantly between groups (W = 25, *p* = 0.83) (Figure 8D). EPI scans are shown for an anesthetized mouse (Figure 8E) and an awake mouse (Figure 8F) in order to illustrate the similarity of image acquisition in awake versus anesthetized mice.

## 4. Discussion

We report here a simple means to create a three-dimensional printed mouse holder for awake-mouse MRI. Within this study, a five-day acclimation paradigm was developed to stabilize and reduce the stress response in animals to the restraints and imaging environment and to limit motion artifacts during image acquisition. Through these means, pristine anatomical images were acquired and showed that anesthesia impacts ^31^P spectroscopy data, MEMRI transport measurements, and rs-fMRI connection signals based on the small-world network index.

The demonstrated drop in cortisol concentration back to baseline in the serum of the experimental mice from day 1 to day 5 has several important implications for the outcomes of our awake functional imaging techniques. Plasma cortisol levels are a commonly used metric to assess stress in mice [31,32,33]. Several studies have used cortisol as an indicator of stress in mice and have observed an increase in cortisol levels as a response to acutely induced stress [34,35,36]. The drop in cortisol levels, shown after the awake imaging acclimation protocol was complete, indicated that the acclimation protocol, and use of an acclimation chamber, was successful in reducing the stress of the mice as they passed through an MRI-like device. This resulted in two major advantages: the first being the reduction of the role of stress as a confounding variable in our review of mouse brain function, and the second being the reduction of movement in the mice during the scanning period. Taken with the standard levels of serum cortisol in naïve mice, our resulting data suggested that during the early stages of the acclimation, mice experienced a drastic increase in cortisol levels. This increase in cortisol levels, corresponding to stress, was expected. After day 5 of acclimation, cortisol levels in the mice decreased, returning to near baseline levels. Although there was not a significant decrease in cortisol levels from day 1 to day 5, the reduction in cortisol levels down to baseline levels indicated that the mice were properly acclimated and had a reduction in stress.

The reduction of head movement in the mice during the awake functional imaging had the clear benefit of reducing background noise in the resultant scans. When compared to non-acclimated mice, the acclimated mice displayed much less head movement, allowing for higher quality scans.

^31^P spectra showed decreased levels of PCr and Pi in the anesthetized mice compared to the awake mice, but no differences in ATP levels were observed. This could be explained by a difference in metabolic flux between awake and anesthetized animals. Assessment of the PCr/ATP ratio in awake and anesthetized mice demonstrated that anesthetized mice had a decreased energy reserve compared to awake mice [37,38]. The Pi/ATP ratio reveals information about the rate of ATP hydrolysis. The larger Pi/ATP ratio in awake relative to anesthetized mice indicated an increase in ATP hydrolysis in the awake animals [37,39]. This finding indicated that there was higher energy consumption in awake mice compared to anesthetized mice. These results are consistent with studies that have shown that the rate of the ATPase reaction, responsible for ATP hydrolysis, decreases in anesthetized animals [40]. Although there was a decrease in the energy reserve, there was a decrease in the consumption of ATP. This accounts for the fact that there was no observed difference in levels of ATP between the two groups animals. The Pi/PCr ratio was decreased in anesthetized mice indicating that awake mice have an increase in brain metabolic activity and mitochondrial function compared to anesthetized mice [37,41]. The significance lies in the fact that bioenergetic impairment, in the forms of brain energy metabolism and mitochondrial function are affected in the neuropathology of certain conditions, such as Schizophrenia and Alzheimer’s disease [42]. When using animal models to study the pathological mechanisms behind certain brain diseases, the use of anesthesia may occlude bioenergetic impairments, which are intrinsic to the exact disease one is hoping to study. For this reason, a translational ^31^P spectroscopy study’s power, particularly for a small study with few animal subjects, could be severely compromised by experimental errors

As revealed by our MEMRI studies, axonal transport rates were decreased by the administration of isoflurane anesthesia, from 0.045 to 0.015 (signal intensity/time), consistent with previous studies that have demonstrated a reduction in axonal transport rates due to anesthesia^21^. The impact of anesthesia towards functional brain imaging, such that the presence of isoflurane anesthesia attenuates measured axonal transport rates in imaged mice, has significant implications for research methods involving MEMRI. For example, certain neurological disorders, such as frontotemporal dementia, result in impaired axonal transport in the substantia nigra [43]. The use of anesthesia and its resulting effect on axonal transport in the whole brain may occlude genuine differences in axonal transport within certain brain regions and significantly affect the translational power of MEMRI studies.

The rs-fMRI data showed a lower small-worldness index for isoflurane anesthetized mice compared to their awake counterparts [awake fMRI small-worldness index = 1.08 +/− 0.05 vs 1.01 +/− 0.01 in anesthetized mice (W = 0, *p* = 0.004)]. This generally higher small-worldness index for awake animals versus anesthetized animals shows the importance of the awake-animal data through evidence that isoflurane reduces the ability to acquire accurate functional connectivity maps. These rs-fMRI data also showed that awake subjects did not produce significantly more motion compared to their anesthetized counterparts; motion for the awake mice was 0.012mm +/− 0.04 compared to 0.003mm +/− 0.003 for the anesthetized mice (W = 13, *p* = 0.792). This motion analysis also supported the view that it was not motion that was causing the observed differences in axonal transport and changes in brain metabolism, as indicated by the ^31^P data.

These studies illustrate the important impact of anesthesia on preclinical MRI studies that assess functional output (metabolism or brain function). Through ^31^P spectroscopy studies, MEMRI studies, and rs-fMRI data, we have presented evidence to support the superiority of awake MRI acquisition for functional studies. However, an important question that should be addressed in awake-animal imaging studies is *“What parameters can be used to define an awake animal?”* In this study, respiration rates were primarily used to monitor when the effects of anesthesia began to diminish and to determine when the mouse was fully awake and ready to be imaged. Under isoflurane anesthesia, the average respiration rate for mice is approximately 55–65 breaths per minute. A fully awake mouse will have an average respiration rate of approximately 160 breaths per minute [18,19,20]. Therefore, the respiration rates of each animal should return to at least 160 breaths per minute before imaging. In this study, after anesthetizing the mice for the purpose of securing them to the holder, we allowed 15 min to elapse between the cessation of low dose (0.5–1%) isoflurane administration and imaging commencement. This period allowed the respiration rate to return to baseline conditions, indicating that the animals were no longer anesthetized, but were rather fully awake and recovered from the anesthesia when we conducted our imaging procedures [18,19,20].

Evidence has shown that suppression of functional connectivity persists after the cessation of isoflurane at a length corresponding to the initial isoflurane anesthetic length and that isoflurane causes suppression of functional connectivity and changes in low-band spectral power [10]. While the functional activity suppression wore off within 1.25 h, the low-band spectra became uniform at about 3.25 h for mice administered with isoflurane for 30 min. Although our mice were anesthetized with 0.5–1% isoflurane for a total of 5 to 10 min before imaging (this variation depended on the time needed to secure each mouse in the holder), this connection between isoflurane and functional activity suppression suggests that the use of isoflurane might have caused longer lasting functional connectivity and spectra changes in our animals, despite them being fully awake by physiological respiration standards. We plan to explore this connection in future studies by taking BOLD rs-fMRI measurements at regular intervals for long periods of time, such as reported by Pan and coworkers [44], to measure functional connectivity suppression and changes in the spectra after isoflurane administration [45]. This study would provide a way to reliably quantify the amount of time to wait to begin accurate awake imaging without lingering traces of functional connectivity suppression and changes in the spectra induced from initial isoflurane administration.

## 5. Conclusions

We have created a novel awake mouse imaging holder, interfaceable with existing Bruker systems, to conduct in vivo preclinical mouse imaging. Here, we made the three-dimensional designs of the holder available for reproduction. Additionally, we developed a five-day acclimation paradigm to stabilize animal stress levels and reduce motion during the preclinical imaging. Through these means, we acquired pristine images and have, for the first time in the preclinical imaging community, demonstrated changes in MRI readouts associated with anesthesia. In order to obtain the most physiologically relevant data, future studies should carefully consider how anesthetics may confound the data and use awake imaging whenever possible. We also validated that rs-fMRI studies should be conducted in awake animals rather than anesthetized animals. We have provided evidence supporting how isoflurane anesthesia administration during imaging lowers brain metabolic activity, as supported by changes in the ^31^P spectra, decreased rates of axonal transport, as evidenced by changes in MEMRI measurements, and decreased rs-fMRI small-worldness index, resulting in inferior functional connectivity measurements.

## Figures and Tables

**Figure 1 biosensors-12-00616-f001:**
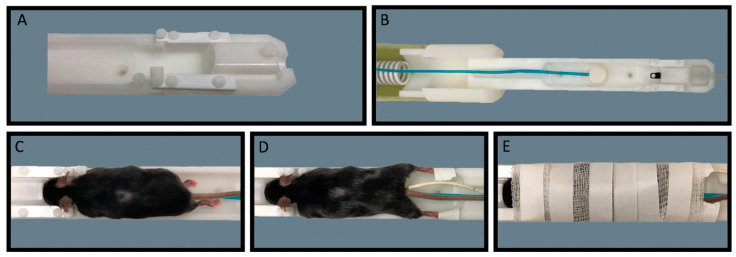
Awake Animal Imaging Holder. A custom mouse holder with nose cone and head brace (**A**), and mouse holder with tooth bar, respiration monitor, and heat source (**B**). Image of mouse placed in holder, with nose in nose cone and on tooth bar (**C**). Image of mouse with limbs taped down and temperature probe inserted (**D**). Image of mouse fully secured with gauze and tape to reduce all movement during imaging (**E**). Oxygen was supplied to the subjects through the oxygen hose during imaging and acclimation.

**Figure 2 biosensors-12-00616-f002:**
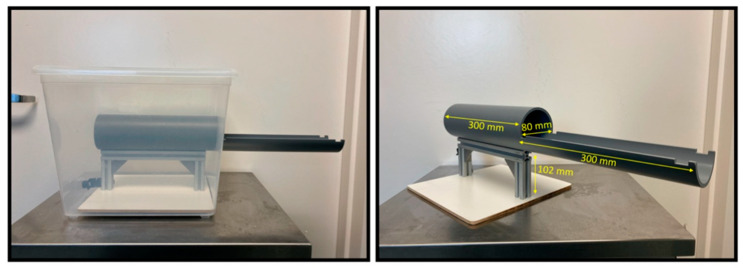
Acclimation Chamber. The simulated acclimation core was cut from PVC pipe and measures 600 mm in total length (300 mm for the core and 300 mm for the open extension). The width of the PVC pipe was 80 mm and the entire core was elevated 102 mm off the platform by a steel rack. The core was designed to mimic a small animal MRI core so that animals acclimated using this structure would exhibit lower stress signals and perform fewer and less drastic head movements during scans.

**Figure 3 biosensors-12-00616-f003:**
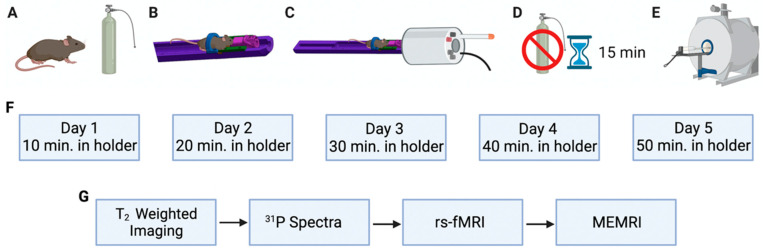
Protocol for 5-day acclimation for awake mouse imaging (**A**–**F**). Protocol for awake image acquisition (**A**–**E**,**G**). Mice were lightly anesthetized under 0.5–1% isoflurane with a flow rate of 2.5 L/minute for approximately 2 to 3 min (**A**). Mice were secured into the holder with a continued delivery of 0.5–1% isoflurane and maintained at 37 °C with an air-heating system (**B**). The mouse holder was slid into coil (**C**), isoflurane was discontinued, and mice were given 15 min to become fully awake (**D**). EPI noises, conduct T_2_ Tripilot, and Anatomy 2b scans were run (**E**). Mice remained in the MRI for set amount of time based on day of acclimation (physiological responses to stress are monitored throughout) (**F**). After Day 5 of acclimation T_2_-weighted imaging was conducted (T_2_ Tripilot position scan was conducted first and animal was adjusted if necessary), ^31^P and T_1_-weighted MEMRI studies were conducted, and rs-fMRI data collected (**G**).

**Figure 4 biosensors-12-00616-f004:**
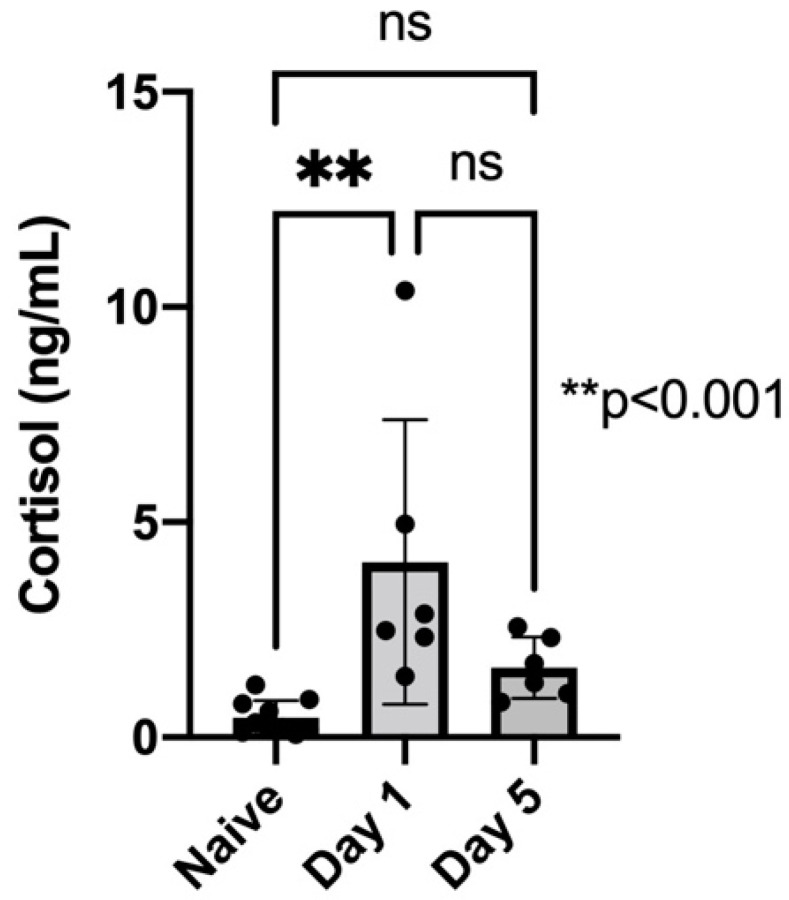
Cortisol levels of naïve (unacclimated) mice (N = 10) compared to cortisol levels of acclimated mice (N = 6) on day 1 and day 5 of the acclimation protocol. Significant increase (*p* = 0.003, one-way ANOVA) in cortisol levels after day 1 of acclimation, compared to naïve mice. After day 5 of acclimation the cortisol levels were no longer significantly different from baseline (naïve) levels.

**Figure 5 biosensors-12-00616-f005:**
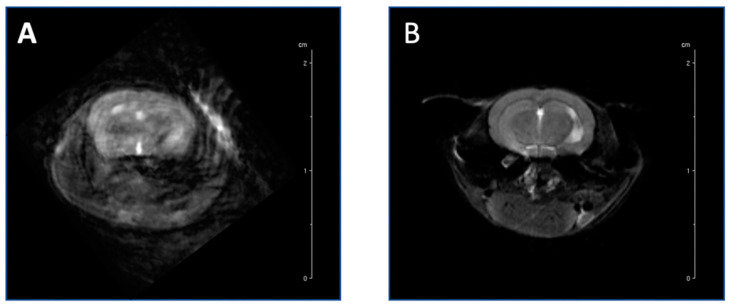
Highly motion-sensitive T_2_-weighted axial images of the brains of two mice show the significant reduction of motion and artifacts during the awake MRI imaging process. Anatomical image acquired with previously designed mouse holder (**A**) vs our newly designed mouse holder (**B**).

**Figure 6 biosensors-12-00616-f006:**
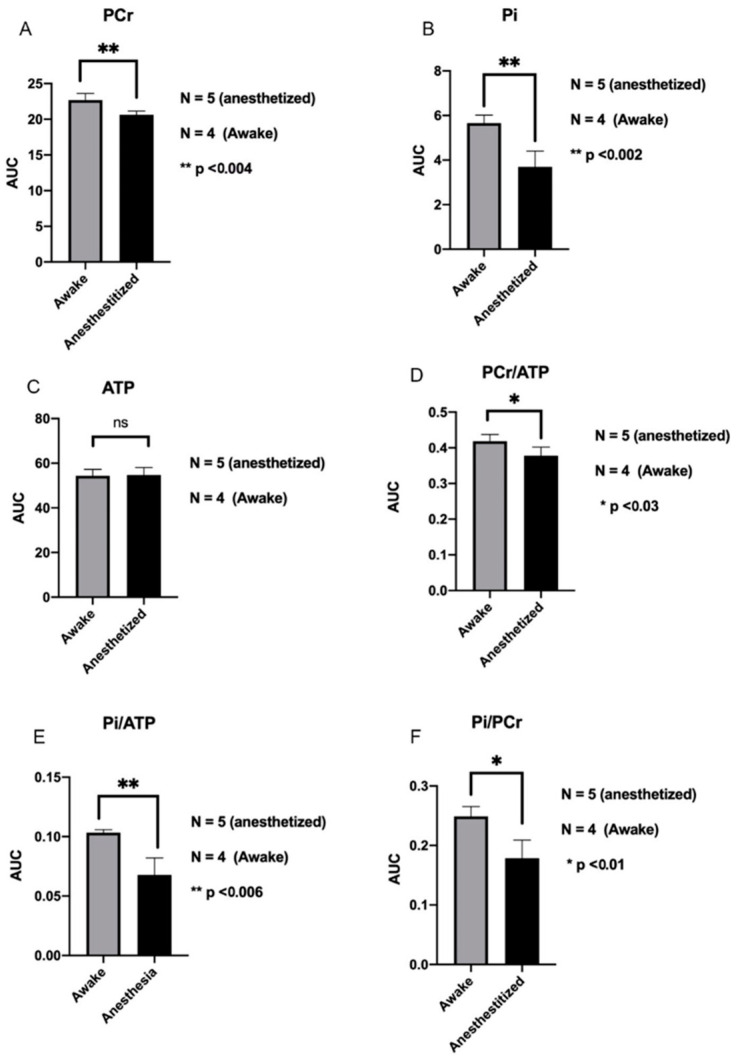
Bioenergetic status of the brains of awake and anesthetized mice. Levels of PCr and Pi decreased in anesthetized mice relative to awake mice (**A**,**B**). ATP levels in the brain were maintained (**C**). PCr/ATP levels in awake mice are increased relative to anesthetized mice (**D**). Increases Pi/ATP levels in awake mice relative to anesthetized mice (**E**). Increased levels of Pi/PCr in awake mice (**F**). N = 5 anesthetized and N = 5 awake mice.

**Figure 7 biosensors-12-00616-f007:**
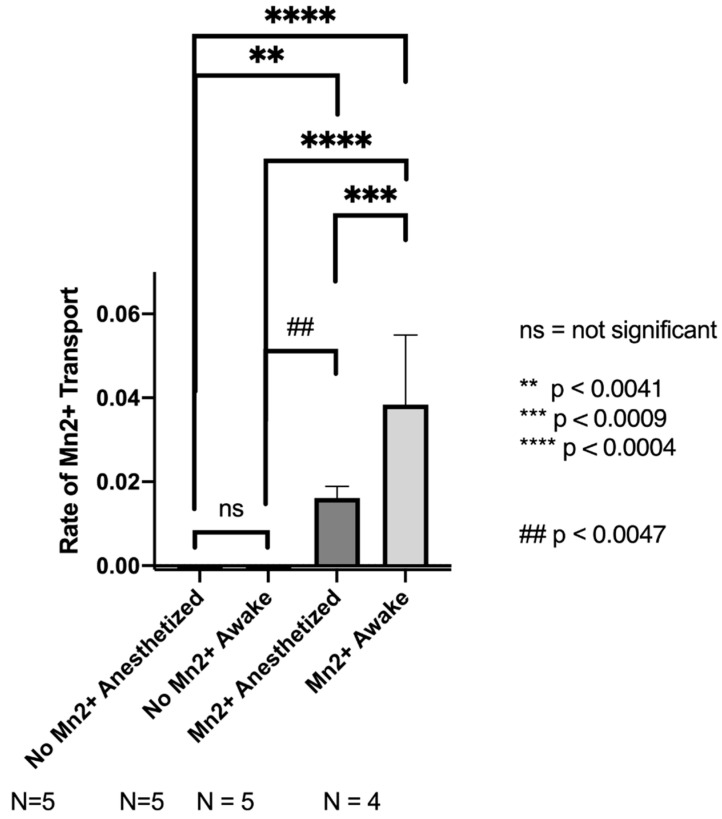
The mean rate of axonal transport of Mn^2+^ (signal intensity/time) in the olfactory bulb of anesthetized animals (0.0161) was significantly lower than the mean rate of axonal transport of Mn^2+^ in the olfactory bulb of awake animals (0.03869). The first two columns labelled “No Mn^2+^ Anesthetized” and “No Mn^2+^Awake” do not contain observable signal because no Mn^2+^was administered to these mice.

**Figure 8 biosensors-12-00616-f008:**
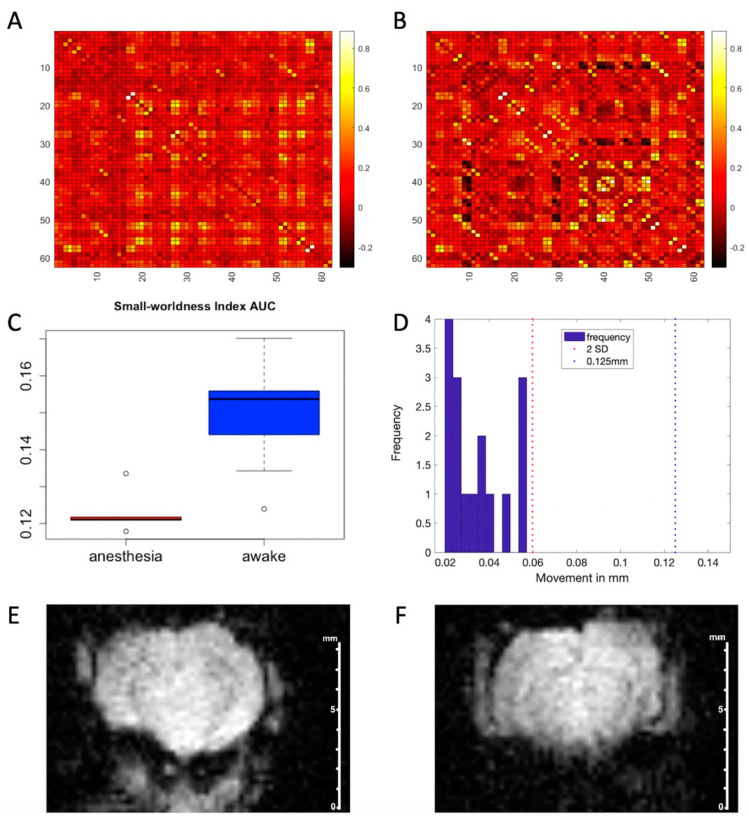
Functional connectivity matrices for anesthetized (**A**) and awake (**B**) mice were highly similar via Mantel test (r = 0.61, *p* < 0.001). See Appendix A for brain regions indicated on X and Y axes. Small-worldness index area under the curve (**C**) was significantly higher in the awake group (W = 1, *p* < 0.001). Movement, as measured by root mean squared framewise displacement (**D**), was below the half in-plane voxel (0.125 mm) and 2 standard deviation (SD) cutoffs for all mice. Motion did not differ significantly between groups (W = 25, *p* = 0.83). EPI scans are shown for an anesthetized mouse (**E**) and an awake mouse (**F**).

## Data Availability

The data that support the findings of this study are available from the corresponding author upon reasonable request.

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
