# Peer review of "A Mouse Holder for Awake Functional Imaging in Unanesthetized Mice: Applications in 31P Spectroscopy, Manganese-Enhanced Magnetic Resonance Imaging Studies, and Resting-State Functional Magnetic Resonance Imaging"

_biosensors, 2022, doi:10.3390/bios12080616_

Round 1

Reviewer 1 Report

In this manuscript the authors have designed an improved 3-D printed awake mouse imaging holder for MRI applications. It has been demonstrated that significant differences are present between anesthetized and awake animals in 31P spectra, MEMRI transport rates, and rs-fMRI connectivity research. This work emphasizes the importance for MRI applications using awake animals. Some revisions are suggested:

1. Line 183-185, I note that the model mice are quite different (age, number) between the group using acclimation protocol in the MRI machine and the group using acclimation protocol in the designed acclimation chamber. Is there any reason for this arrangement?

2. Line 393-395, the authors demonstrated “the reduction of motion artifacts when using the newly designed restraint and new acclimation procedures compared to the original holder.” So, can they provide some quantitative results to better confirm this conclusion?

3. Figure 6, I wonder whether the significant difference between awake and anesthetized group is solid due to the limited number (4~5) of samples.

4. The resolution for several images may be too low, especially Figure 8, which should be improved.

5. The citation style should be uniform. For example, [17], (Stenroos et al., 2018), superscript 17, are all present in the context.

Reviewer 2 Report

This work proposes a mouse holder that is interfaceable with existing MRI systems and enables awake in vivo mouse imaging. The authors have demonstrated significant differences in 31P spectra, MEMRI transport rates, and rs-fMRI connectivity between anesthetized and awake animals, which may offer real functional studies in animal models. The manuscript can be recommended for publication after revisions: 1) I cannot clearly find the advantages of the proposed mouse holder over those used in existing MRI systems. Can the authors provide some data obtained by “previously designed holders” for comparison? 2) Since all four feet and the tail of mice are taped down in the current protocol, I wonder whether this may activate any stress response. Have the authors checked the changes in levels of dopamine or serotonin? 3) Fig. 7, what is the unit for rate of Mn2+ transport? Also in this experiment, the number for mice in each group are not same, with only 4 mice in “Mn2+ awake” group, but 5 mice in other groups. 4) Figure 8A, B and C, what do X and/or Y axis mean? The scale bar is not clear in Figure 8E and F. 5) The introduction part is generally long and can be truncated. I think it is unnecessary for the authors to put so detailed results (Line 104- Line 138) before the main text are exhibited.

Round 2

Reviewer 1 Report

In this manuscript, the authors describe a newly designed mouse holder that is interfaceable with existing MRI systems and enables awake in vivo mouse imaging. 31P spectra, MEMRI transport rates, and rs-fMRI connectivity studies demonstrated significant differences between anesthetized and awake animals. This study hopes to use awake animals for MRI applications to reduce the impact of anesthetics on the diagnosis of special model animals, and the research has certain practical significance. Some revisions are suggested:

1. Does the Mouse Holder have requirements on the size of the mice (the weight of mice varies greatly by age)? What are the upper and lower limits of mouse volume? The authors should provide the relevant experimental data.

2. The stress response of animals generally changes the content of several hormones, such as adrenocorticotropic hormone, glucocorticoid, angiotensin, etc. The authors only tested glucocorticoids as one indicator, and whether other hormone changes were also detected.The authors should provide these experimental data.

3. In 31P Spectra Studies, the authors pointed out that the ratio of Pi/PCr decreased in anesthetized mice, indicating decreased metabolic activity and mitochondrial function (Fig. 6F), but ATP levels remained unchanged (Fig. 6C). The authors speculate that it may be a change in metabolic flux in anesthetized mice. Is there a clear research report on this speculation? What factors influence the changes in metabolic flux in this experiment?

4. The authors point out that imaging in awake mice reduces the effect of anesthetics on the diagnosis of some mouse models of brain metabolic or neurological diseases, and has there been any attempt to conduct experiments in these mouse models.The authors should add these experimental data.
